# Chiral Selectivities of Permethylated α-, β-, and γ-Cyclodextrins Containing Gas Chromatographic Stationary Phases towards Ibuprofen and Its Derivatives

**DOI:** 10.3390/ijms25147802

**Published:** 2024-07-16

**Authors:** Zoltan Juvancz, Rita Bodane-Kendrovics, Csaba Agoston, Balazs Czegledi, Zoltan Kaleta, Laszlo Jicsinszky, Gergo Riszter

**Affiliations:** 1Rejtő Sándor Faculty of Light Industry and Environmental Engineering, Institute of Environmental Engineering and Natural Science, Óbuda University, Doberdó út 6, H-1034 Budapest, Hungary; bodane.rita@uni-obuda.hu (R.B.-K.); agoston.csaba@uni-obuda.hu (C.A.); 2Department of Organic Chemistry, Semmelweis University, Hőgyes Endre Street 7, H-1092 Budapest, Hungary; czegledi.balazs@stud.semmelweis.hu (B.C.); riszter.gergo@phd.semmelweis.hu (G.R.); 3Dipartimento di Scienza e Tecnologia del Farmaco, University of Turin, Via P. Giuria, 9, 10125 Turin, Italy; laszlo.jicsinszky@unito.it; 4Artificial Transporters Research Group, Institute of Materials and Environmental Chemistry, Research Centre for Natural Sciences, Eötvös Loránd Research Network, Magyar Tudósok Körútja 2, H-1117, Budapest, Hungary

**Keywords:** gas chromatographic chiral separations, permethylated cyclodextrin selectors, ibuprofen, structure-chiral selectivity relationships, molecular modeling

## Abstract

Ibuprofen is a well-known and broadly used, nonsteroidal anti-inflammatory and painkiller medicine. Ibuprofen is a chiral compound, and its two isomers have different biological effects, therefore, their chiral separation is necessary. Ibuprofen and its derivatives were used as model compounds to establish transportable structure chiral selectivity relationships. Chiral selectors were permethylated α-, β-, and γ-cyclodextrins containing gas chromatographic stationary phases. The chiral selectivity of ibuprofen as a free acid and its various alkyl esters (methyl, ethyl, propyl, isopropyl, butyl, isobutyl, and isoamyl esters) derivatives were tested at different temperatures. Every tested stationary phase was capable of the chiral separations of ibuprofen in its free acid form. The less strong included *S* optical isomers eluted before *R* optical isomers in every separate case. The results offer to draw transportable guidelines for the chiral selectivity vs. analyte structures. It was recognized that the *S* isomers of free ibuprofen acid showed an overloading phenomenon, but the *R* isomer did not. The results were supported by molecular modeling studies.

## 1. Introduction

Enantiomers, optical isomers, are asymmetric molecules which cannot superimpose with their mirror images [1]. The members of an enantiomeric pair can show rather different biological effects, despite their very similar structures [2]. The possible difference in the biological effects of the members of enantiomeric pairs have forced the authorities to introduce directives to use only optically pure pharmaceutical products [3]. The enantiomer-pure products make chiral selective productions necessary [1], or to separate one member of the enantiomeric pair [4]. Enantiomer selective analysis is also important in their quality control and metabolite studies [5].

Enantiomer selective separations are still very challenging tasks in chromatography [6]. The members of enantiomeric pairs are indistinguishable in the homogeneous space, but they can be distinguished from each other in inhomogeneous surroundings. The chiral separation of enantiomers requires asymmetric chiral separation agents, which produce inhomogeneity space in their interactions. The overwhelming parts of the chromatographic enantiomer separations are based on the three-point interactions model [7]. According to the simplified model, one isomer interacts simultaneously with two interaction points of the selector. The other isomer interacts simultaneously at three points with the selectors. The three-point interactions have bigger interaction energies than isomers with only two interaction points. The three-point interactions require a perfect fit between the selector and selectand in their chemistry features (e.g., H donor–acceptor, π-π, acid–base interactions) and a perfect fit in the steric arrangement of the interacting groups. A chiral selective agent has to produce a perfect fit with only a limited number of enantiomer pairs. No universal chiral selector exists because the enantiomers with different sizes and chemical characters need tailor-made chiral selective agents. An efficient chromatographic system can separate enantiomeric pairs, having only 0.1 kJ/mol interaction energy difference with the separation agent [8].

Cyclodextrins (CD), cyclic oligosaccharide molecules [9], are frequently used as chiral selectors in chromatography [10,11].

The CDs have very broad chiral selectivity spectra for the following reasons [10]:
The CDs have numerous chiral centers (35 in β-CD). These chiral centers differ from each other. The CDs have twisted truncated shapes, resulting in different arrangements (lengths, directions) of the functional groups around every chiral center. This is the reason why the CDs have broader chiral recognition spectra than linear amylose molecules, which have the same arrangements in every glucose unit.The added functional groups of derivatized CDs (phosphate, sulfate, amino, naphthyl, acetate) offer further interaction abilities and steric arrangements to selectand enantiomers compared to native CDs.The majority of CD derivatives are randomly substituted molecules. They are not uniform products. They differ from each other in their numbers and in the positions of the substituents, resulting in every isomer having a different chiral recognition feature.The derivatized CDs have a rather flexible structure. They can change their shapes to interact intimately (more strongly) with functional groups of analytes with the so-called “induced fit” mechanism. The “induced fit” phenomenon further increases the chiral selectivity spectra of CDs.The ionizable CDs can broaden their selectivity spectra according to their ionization states.The types of mobile phases or background buffers influence the selectivity features of CDs. The CDs can simultaneously include not only the analytes but also a solvent molecule in their cavities.

Generally, the CDs favor the alpha positions of interacting groups of analytes from an asymmetric center. Such positions have increased the rigidity of the analyte around the asymmetric center. However, they can also separate analytes with interacting groups at beta, gamma, and other positions from their chiral centers. CDs can also separate enantiomers having planar or axial chirality, not only enantiomers with central chirality. Moreover, CDs are good chiral selectors toward enantiomers having heteroatomic chiral centers.

The great variety of chiral centers and induced fit can give multimodal characteristics to the CDs. The multimodal character can offer more than one chiral recognition mechanism for certain analytes. The inclusion phenomenon can be a key interaction in the chiral recognition of CDs, but the inclusion is not necessary for their chiral recognition.

The cyclodextrins are the chiral selectors in cases of chromatographic techniques that use capillary columns [10,11]. The very high performance of these techniques compensates for the moderate selectivity of cyclodextrins. The cyclodextrins show moderate selectivities, which come from their flexibility.

Because of their multimodal characteristics and relatively low chiral selectivity, CDs’ chiral recognition capabilities make it challenging to forecast the result of a chiral resolution for a particular enantiomer pair. Therefore, the majority of the successful chiral separations are the results of trial-and-error development methods. However, systematic chromatographic studies can establish guidelines for chiral selectivity mechanisms of cyclodextrins using model compounds [12,13]. These guidelines are useful in the method development of other enantiomeric pairs.

The up-to-date scientific results contain modeling studies [14,15]. The results of chromatographic investigations support frequent molecular modeling [16,17,18,19,20] or X-ray [21,22], IR [23], UV [24], and NMR data [16,18].

In this paper, ibuprofen (Figure 1) was chosen as a model compound to obtain a deeper knowledge of the chiral selectivity vs. structure of analytes on permethylated CD selectors.

Ibuprofen is a nonsteroidal anti-inflammatory drug (NSAID), a very popular over-the-counter medicine under various trade names and formulations [23]. The *S*-enantiomer of ibuprofen is believed to be the more pharmacologically active enantiomer, but it is known that the *R*-enantiomer undergoes extensive interconversion to the *S*-enantiomer in vivo. Ibuprofen is frequently administered as a racemic mixture, however, the enantiomerically pure *S* product (Dexibuprofen) has a significant market share too.

Ibuprofen was already chirally separated with various chromatographic techniques using different separation agents. Gas chromatography is an appropriate technique for the separation of ibuprofen enantiomers with CD-containing stationary phases [24]. The packed column SFC regularly uses polymeric chiral stationary phases for the separation of ibuprofen enantiomers [25]. On the other hand, CD-containing polymers were also applied in capillary column SFC [26]. The HPLC makes regular chiral separations of ibuprofen with cellulose and amylose derivatives chiral stationary phases [27]. CD derivative was also appropriate as a chiral mobile phase additive [28]. The various cyclodextrin derivatives are overwhelmingly used for the separation of ibuprofen isomers in the practice of capillary electrophoresis [29,30].

In this study, trimethylated α-, β-, and γ-CDs containing gas chromatographic phases were tested to figure out the role of cavity sizes of CD in chiral separations. The tested compounds were the ibuprofen itself and its different alkyl esters. The differences in chiral selectivities between the free acid and alkyl derivatives demonstrate well the role of hydrogen bonding in the chiral recognition features. The selectivity differences among the various alkyl esters give information about the role of inclusion in chiral separations.

The systematically gained data allow guidelines to be established for the chiral separation mechanism of CDs, which are transferable to other enantiomer pairs.

## 2. Results

In several cases, chiral separations of ibuprofen enantiomers were achieved, which are summarized in Table 1. First, the elution orders of the *S* isomer were observed for every separate test material.

Kováts retention indices [32] (KI) were added to Table 1 because these values showed a better relationship between the retention of various tested compounds than retention times measured on different stationary phases. The calculated boiling points [31] of the tested materials were also added to Table 1. The increased KI values of tested compounds refer to the inclusion of these materials, comparing their boiling points.

The chiral selectivities were not recognized in several cases, which were assigned with <1.01 symbols. Even 260 min were required to recognize some chiral selectivity on some occasions, as happened on gamma DEX for the isobutyl ester of ibuprofen. Where measurable chiral selectivities were achieved at temperatures higher than 100 °C, the retention parameters were calculated for 100 °C. It was possible because the ln α − 1/T relations showed linear characters in a broad temperature range (Figure 2).

The free acid could be baseline separated (α > 1.024) with alfa-, beta-, and gamma-permethylated CD containing stationary phases. The most effective chiral recognition was 1.129 on β-CD containing stationary phase at 100 °C. The chiral selectivity of the α-CD and γ-CD containing stationary phases showed much lower chiral selectivity—1.027 and 1.029 at 100 °C, respectively. Their representative chromatograms are in Appendix A (Appendix A).

The KI values of free acids were significantly higher (KI: α-CD, 2033; γ-CD, 2043) on alfa and gamma selectors than the same parameter on the beta selector (β-CD, 1906). These observations suggest that the α-CD and γ-CD selectors have intensive inclusion complexations towards the free acid of ibuprofen. These observations suggest that their KI values are higher than the KI values of their isoamyl derivatives (KI: α-CD, 1953; γ-CD, 1940), despite the isoamyl derivative of ibuprofen having a higher boiling point (330 °C) than the boiling point of the free acid of ibuprofen (323 °C). On the other hand, the beta selector produced a lower KI value with free acid (1906) than its isoamyl derivative (1915).

An interesting phenomenon was observed when analyzing the free acid of ibuprofen on the β-CD-based selector. The early eluting *S* isomers produced significant peaks overloading in higher concentration ranges (Figure 3A). The *S* isomers had a fronting peak profile with approximately 20% efficiency loss compared to the *R* isomers. Probably the members of the ibuprofen enantiomer pair belong to different types of interactions. The peak shape of the *S* isomers also became normal in a one-magnitude-less concentration range (Figure 3B).

The overloading of the *S* isomer was a consequence of the limited capacity of inclusion interactions of this isomer, compared to the *R* isomers. The *R* isomer might be included in another way (higher capacity) or connected to the outer sphere of permethylated β-CD. The limited capacity of the included enantiomers has already been mentioned [33]. Probably this limited loading capacity of *S* isomer comes from the strict steric requirements between the host and guest partners in the tightly included complexes. The cited paper did not refer to loading capacity differences among the members of enantiomer pairs. The linearity of the ln α − 1/T relationship was possible because both chiral recognition mechanisms favored the *R* isomer. The different peak asymmetry of the members of the enantiomers was also mentioned in another source [34].

Only the ibuprofen methyl ester was baseline separated from the ester derivatives on the permethyl β-CD containing selector which is shown in Appendix A (Appendix A). Some other ester derivatives were also partially chiral separated on different selectors according to Table 1, which are presented in the Appendix A too (Appendix A).

Molecular modeling studies started from the crystalline complex of *S*-ibuprofen with permethylated β-CD [20]. The published structure originated from aqueous crystallization, which can significantly influence both the crystal structure and the location of guest molecules in the CD cavity [35]. The molecular modeling experiments also used the complexes of the ibuprofen esters in reversed orientation compared to the crystal structure of permethylated β-CD and *S*-ibuprofen. Our simulations showed that calculated energy relationships of the molecular mechanics and dynamics methods [36] are comparable with the semiempirical [37] quantum chemical calculations (to be publish). The favorable energy changes from the separated and complexed states were in the 15–25 kcal/mol range, albeit the observed energy differences were usually insignificant. The molecular dynamics studies also revealed that the guest molecules had several closely equal low energy states inside or close to the cavity. The flat energy surfaces also pointed out the effect of the secondary forces in the molecular associations. The molecular shape of the permethylated β-CD often showed significant distortions, but the energy gain of intermolecular associations could compensate for the stressed CD structure. It is also true that nearly 80–85% of the molecular association energies contributed by the CD, and the energies of CD structure distortions (e.g., conformational energy changes), were commensurable with the energy gain of the intermolecular interaction energies. Generally, the *S*-isomer favored the inclusion of complex associations, while the R-isomers showed stronger intermolecular interactions between the pendant groups of CD and ibuprofen esters, although not always. Unfortunately, the intermolecular energy differences between the *R*- and *S*-isomer and the CD were rarely near the significance limit. These limits in molecular mechanics-based cases were 5 kcal/mol and 3 kcal/mol in the quantum chemical method. The Monte-Carlo global minimum energy search pointed out that, despite the entire inclusion of the ibuprofen derivatives among the lowest energy states, those are not necessarily the lowest ones, and both the molecular mechanics energy differences and the heat of formations (from the semiempirical calculations) were minimal. Considering the various arrangements of the guest molecules within the identical complexes, the averaged energy variances were even lower and occasionally had perplexing values. These effects compelled us to not only simulate the fully relaxed structures and their energies but also to simulate the behavior of complexes at different temperatures. The preliminary results with the permethylated β-CD (and calculations with the permethylated α- and γ-CDs are in progress) complexes showed good agreement with the experimental data, as at lower temperatures (e.g., at 100 °C), the dissociation of complexes was significantly slower than at 170 or 200 °C.

Interestingly, in almost all cases of ibuprofen ester-reversed orientation, the S-isomer had the lowest heat of formation, and the intermolecular interaction energies also generally favored the S-isomer. These findings pointed out that, in the case of peralkylated CDs, the former hydroxyl rims became more lipophilic than the cavity, confirming that, in the case of unsubstituted CDs, the cavities are only less hydrophilic than the hydroxyl rims but not hydrophobic. The calculations also supported that, generally, the isoalkyl chain likes the cavity better than the normal alkyl, at least in the case of permethylated β-CD.

The optimized structures of various ibuprofen-containing complexes showed that the interactions of functional groups near the chiral center, the methyl and carbonyl oxygen, and the methoxy moieties of the permethylated CD, are crucial for chiral recognition. The effect of the chiral methyl moiety has a significantly smaller influence on the separation of the unsubstituted acids since the involvement of the acid proton in the formation of H-bonds between different oxygen species, including glucosidic ones, is the main separating force. According to textbooks, the distance between donor and acceptor atoms in H-bonds is usually 2.7–3.3 Ǻ. Typically, these distances were close to the lower limit for esters, as shown in Figure 4A. Since quantum–chemically optimized structures have the optimum energy at 0 K, much higher temperatures, ~400–500 K, are used in GC experiments. At experimental temperatures, the flexibility of the molecules strongly influences the conformations and interactions between molecules. Data from ongoing MD simulations will provide deeper insight into the formation of intermolecular interactions of non-relaxed molecular assemblies and their impact on the efficiency of chiral separation.

The hydrophobic interactions (van der Waals forces) prevail at slightly larger carbon–carbon distances, usually ~3.3–4 Ǻ. The interaction of chiral methyl groups with the O-methyl of permethylated CD was generally within this distance in the optimized structures. We found at least three different main methyl–methyl interactions between the molecules. The shortest distances between the methyl groups are in Figure 4A. The chiral methyl groups can interact:(a)With O(2)CH_3_ and O(3)CH_3_ units of the same glucopyranoside unit;(b)With O(2)CH_3_ and O(3)CH_3_ units of adjacent glucopyranoside units;(c)With different glucopyranoside units of the macrocycle O(2)CH_3_ and O(2)CH_3_, or O(3)CH_3_ and O(3)CH_3_ separated by two or three units.

The interactions (shorter CH_3_⋯CH_3_ and H-bond distances) often moved in parallel, indicating that both H-bonds and hydrophobic interactions are essential for chiral separation. The weak hydrophobic interactions for both R- and S-ibuprofen also confirm that, not surprisingly, CO⋯HC bonds are significantly weaker than COOH⋯O H-bonds and the missing OH groups push hydrophobic interactions to the forefront. The values of the shortest distances between the weak H-bonds and the methyl groups are close to the limit, which may explain the weak chiral selectivity. In these cases, MD simulations will help to understand the effect of higher experimental temperatures than the optimized structures.

Analysis of the geometric center distances of permethylated CD and ibuprofen derivatives showed that longer straight-chain esters move the entire guest out of the confined state, mainly due to the interaction of the ester alkyl chain with the methoxy groups. Nevertheless, this analysis is less sensitive because the movement of the geometric centers of the guest rarely follows the same direction in different complexes. The shortest-branched alkyl, the i-propyl moiety, was closest to the host center, and the i-pentyl groups immersed nearly the same as n-propyl or n-butyl. The longest distance among the i-alkyl esters studied was in the case of i-butyl, showing interactions between the two terminal methyl moieties and the methoxy groups of the macrocycle (Figure 4B). The alkyl groups in the reversed guest orientation could more easily enter the CD cavity, albeit the host–guest interaction with the primary OCH_3_ groups further complicated the molecular energies.

The alkyl ester derivatives of ibuprofen showed much lower chiral selectivity than the free acid of ibuprofen. Only the methyl ester derivative of ibuprofen showed baseline separation on β-CD containing stationary phase achieving α value 1.027 at 100 °C. However, other alkyl derivatives showed minor or no chiral selectivity on any of the tested selectors. For example, the peaks of the ethyl ester ibuprofen enantiomers showed a definite peak, broadening even at 90 °C, but not splitting. The tendency seems to be that the branched alkyl esters produced a somewhat higher selectivity than normal alkyl esters. According to the literature, the branched alkyl chains fit better into the cavity of β-CD than the normal alkyl derivatives [9]. The low chiral recognition features of alkyl esters appear to be due to their lack of hydrogen donor ability.

## 3. Materials and Methods

The racemic ibuprofen was a product of Merck (Merck LifeScience Kft, Budapest, Hungary), and the *S* ibuprofen was donated by the Department of Organic Chemistry and Technology, Budapest University of Technology and Economics. The following solvents and reagents used were the products of Merck (Merck LifeScience Kft, Budapest, Hungary): hexane, ethyl acetate, methanol, ethanol, n-propanol, i-propanol, n-butanol, i-butanol, i-amyl alcohol, NaOH, and cc. HCl. The chiral selective stationary phases used were the following: Alpha Dex (30 m × 0.25 mm, d_f_ 0.25 μm) mixture of silicone polymer and permethyl α-CD (Merck LifeScience Kft, Budapest, Hungary); Cydex-B column (25 m × 0.22 mm, d_f_ 0.25 μm) mixture of silicone polymer and permethyl β-CD, (SGE, Melbourne, Australia); Gamma Dex (30 m × 0.25 mm, d_f_ 0.25 μm) mixture of silicone polymer and permethyl γ-CD (Merck LifeScience Kft, Budapest, Hungary). The series of alkanes (tetradecane, pentadecane, and hexadecane) were the product of Reanal (Budapest, Hungary).

GC/MS instruments (GC17A-Qp 5000, Shimadzu, Kyoto, Japan) were used for the measurements.

Ibuprofen esters were prepared according to a procedure reported in the literature [38]. To a 0 °C solution of ibuprofen in the appropriate alcohol, thionyl chloride (~4 eq in each case) was added dropwise. After the addition had been completed, the reaction was warmed to room temperature and stirred overnight. The solvent was removed under reduced pressure carefully. The resultant material was dissolved in ethyl acetate, washed with sodium bicarbonate and brine, dried over sodium sulfate, and evaporated to dryness. The resulted crude ester was used as received.

The separation of every tested molecule was attempted at 3 different temperatures at least. The measured retention times were used to calculate the chiral selectivity values and their Kováts retention indices (KI) at 100 °C [32]. The calculations were based on the linearity of the natural logarithm of chiral selectivity (relative retention of members of enantiomeric pair) values of tested materials (lnα) in function of the reciprocal value of absolute temperature values (1/T) [8]. The retention times (100 °C) of tetradecane, pentadecane, and hexadecane were the basis of the calculation of KI values.

Regularly, the tested mixtures had *S* isomer excess to establish the retention orders of isomers.

## 4. Conclusions

The free acid of ibuprofen was chiral separated on every tested column. Most likely, the H-donor abilities of the free acids played an important role in their chiral recognition. The alkyl derivatives of ibuprofen showed little or no chiral selectivity, lacking H-donor ability. The β-CD containing stationary phase was better than α-CD and γ-CD containing stationary phases for the separation of ibuprofen and its alkyl derivatives. An interesting phenomenon was also recognized. The *S* isomer of ibuprofen showed unusual overloading effects on β-CD containing selector.

Although the molecular modeling studies in the recent state could somehow confirm the idea of chiral separation based on the experimental data, further molecular dynamics studies at various temperatures could provide a deeper insight into the behavior of ibuprofen complexes. The reversed orientation of the ibuprofen esters can be more stable than the original ones. This finding may be associated with a higher number of secondary interactions. In the case of acid, the glycosidic oxygens can form stronger H-bonds than the methyl ether oxygens. In addition, the lone pairs of the carboxylic acid oxygen esters can also interact better with the lone pairs of the CD oxygens. Ongoing molecular modeling experiments may provide deeper insights into understanding non-covalent enantioseparation mechanisms.

## Figures and Tables

**Figure 1 ijms-25-07802-f001:**
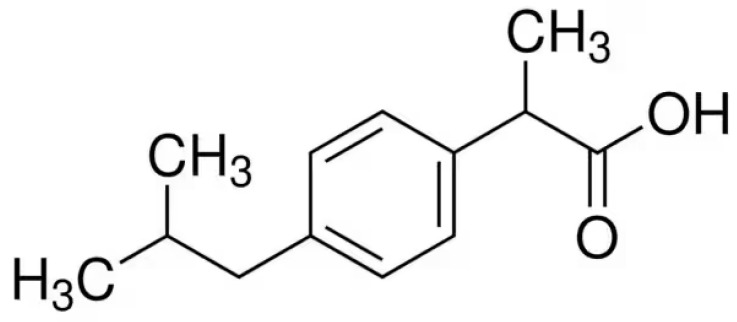
Chemical structure of ibuprofen ((*RS*)-2-(4-(2-methylpropyl)phenyl)propanoic acid).

**Figure 2 ijms-25-07802-f002:**
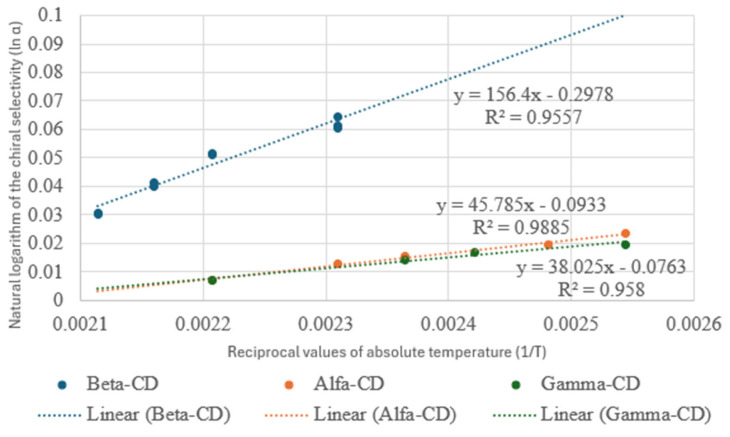
The natural logarithm of the chiral selectivity (ln α) values of tested compounds are represented in the function of reciprocal values of absolute temperature (1/T). The linearity of the curves shows that the chiral separation mechanisms did not change at different temperatures.

**Figure 3 ijms-25-07802-f003:**
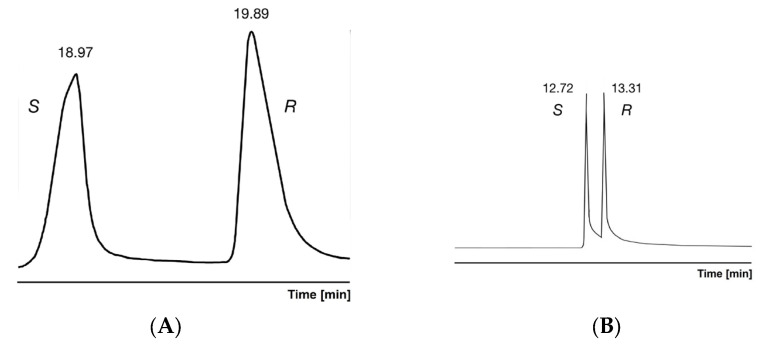
(**A**) Chiral analysis of free acid ibuprofen on permethylated β-CD containing stationary phases. Conditions: instrument, Shimadzu QP5000 GC/MS; column, 25 m × 0.22 mm FSOT; stationary phase, Cydex-B (0.25 μm); carrier, He (50 cm/s); analysis temperature, 170 °C; solvent concentration 2 mg/mL. (**B**) Chiral analysis of free acid ibuprofen on permethylated β-CD containing stationary phases. Conditions: instrument, Shimadzu QP5000 GC/MS; column, 25 m × 0.22 mm FSOT; stationary phase, Cydex-B (0.25 μm); carrier, He (50 cm/s); analysis temperature, 180 °C; solvent concentration 0.2 mg/mL [24]. Symbols. *S*, *S*-optical isomer; *R*, *R*-optical isomer.

**Figure 4 ijms-25-07802-f004:**
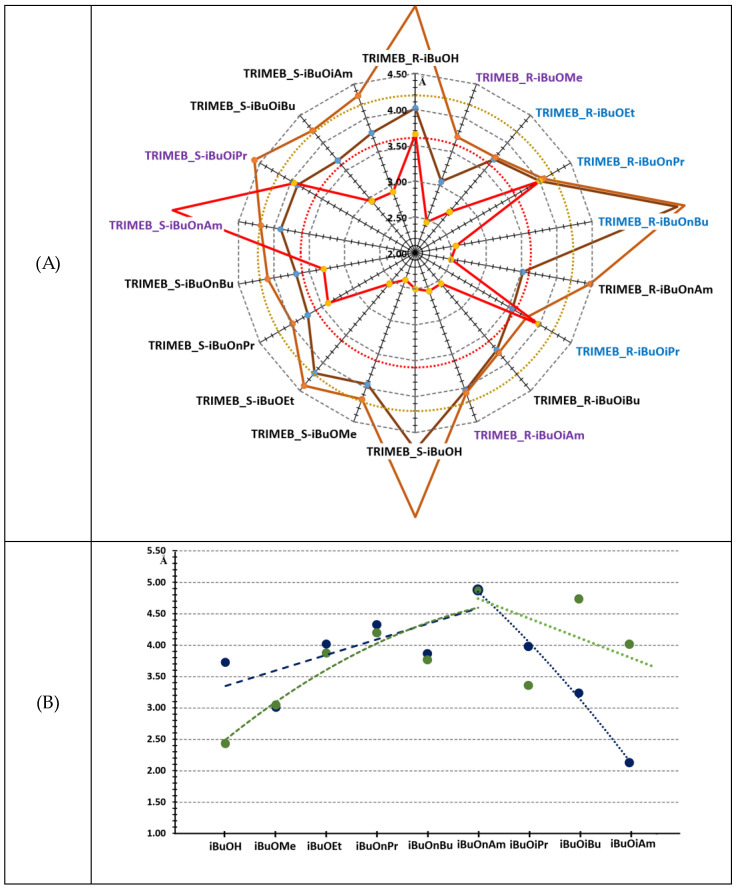
(**A**) Radar diagram of C=O…H and CH_3_…CH_3_ distances (red: C=O…H; brown: shortest CH_3_…CH_3_; dark yellow: second shortest CH_3_…CH_3_ distances). Black text: O(2)CH_3_ and O(3)CH_3_ units of the same glucopyranoside unit; blue text O(2)CH_3_ and O(3)CH_3_ units of adjacent glucopyranoside units; violet text: different glucopyranoside units of the macrocycle O(2)CH_3_ and O(2)CH_3_ or O(3)CH_3_ and O(3)CH_3_. Dashed rings are the upper limit of interaction distances (red: H-bond; dark yellow: hydrophobic). (**B**) Distances of permethylated β-CD and guest molecule geometric centers (blue: R-ibuprofen; green: S-ibuprofen derivatives). Dashed lines show trends. Symbols: TRIMEB, trimethylated β-cyclodextrin; R-iBuOH, (*R*)-ibuprofen (free acid); *R*-iBuOMe, (*R*)-ibuprofen methyl ester; R-iBuOEt, (*R*)-ibuprofen ethyl ester; R-iBuOnPr, (*R*)-ibuprofen n-propyl ester; R-iBuOnBu, (*R*)-ibuprofen n-butyl ester; R-iBuOnAm, (*R*)-ibuprofen n-amyl ester; R-iBuOiPr, (*R*)-ibuprofen isopropyl ester; R-iBuOiBu, (*R*)-ibuprofen isobutyl ester; R-iBuOiAm, (*R*)-ibuprofen isoamyl ester; S-iBuOH, (*S*)-ibuprofen (free acid); S-iBuOMe, (*S*)-ibuprofen methyl ester; S-iBuOEt, (*S*)-ibuprofen ethyl ester; S-iBuOnPr, (*S*)-ibuprofen n-propyl ester; S-iBuOnBu, (*S*)-ibuprofen n-butyl ester; S-iBuOnAm, (*S*)-ibuprofen n-amyl ester; S-iBuOiPr, (*S*)-ibuprofen isopropyl ester; S-iBuOiBu, (*S*)-ibuprofen isobutyl ester; S-iBuOiAm, (*S*)-ibuprofen isoamyl ester.

**Table 1 ijms-25-07802-t001:** The achieved and calculated chiral selectivity values of ibuprofen and its derivatives in different permethylated cyclodextrin containing stationary phases at 100 °C.

Compounds	Chiral Selector						Boiling Point
	Permethylated α-CD		Permethylated β-CD		Permethylated γ-CD		°C, 760 Hgmm [31]
	alfa *	KI **	alfa *	KI **	alfa *	KI **	
Underivatized ibuprofen	1.027	2033	1.129	1906	1.029	2043	323
Ibuprofen methyl ester	<1.01	1686	1.027	1652	<1.01	1717	282
Ibuprofen ethyl ester	<1.01	1738	~1.01	1696	<1.01	1762	297
Ibuprofen isopropyl ester	<1.01	1724	1.013	1703	<1.01	1760	303
Ibuprofen propyl ester	<1.01	1786	1.11	1782	<1.01	1845	312
Ibuprofen isobutyl ester	<1.01	1821	<1.01	1825	1.012	1881	317
Ibuprofen butyl ester	<1.01	1913	<1.01	1866	<1.01	1931	325
Ibuprofen isoamyl ester	<1.01	1953	<1.01	1915	<1.01	1940	330

* Chiral selectivity at 100 °C, ** Kováts retention index [32] (KI) at 100 °C (the mean values of retention times were the basis of calculations of KIs where chiral separations were achieved).

## Data Availability

The data presented in this study are available on request from the corresponding author.

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
