# Peer review of "Chiral Selectivities of Permethylated α-, β-, and γ-Cyclodextrins Containing Gas Chromatographic Stationary Phases towards Ibuprofen and Its Derivatives"

_ijms, 2024, doi:10.3390/ijms25147802_

Round 1

Reviewer 1 Report

Comments and Suggestions for Authors

The authors report a study of ibuprofen and its alkyl esters (methyl, ethyl, propyl, isopropyl, butyl, isobutyl, and isoamyl esters) derivatives, where they were used as model compounds to establish transportable structure chiral selectivity relationships in permethylated α-, β-, and γ-cyclodextrins as gas chromatographic stationary phases at different temperatures. According to their results stationary phase was capable of the chiral separations of ibuprofen in its free acid form, and shown the less strongly S isomers eluted before R isomers in every case. As well the results were supported by molecular modeling studies.

However, before accepting its publication, I consider that the authors should address the following comments:

1) It would be advisable to include figures that illustrate the results of molecular modeling to highlight the possible interactions of the two enantiomers of ibuprofen with cyclodextrins.

2) Also, make a greater discussion in relation to the alkyl groups of the esters that explains the poor recognition of these with the cyclodextrins.

3) Include in the supplementary material the chromatograms that were obtained for each of the racemic esters in their alkyl ester derivatives.

4) Minor fixes:

a) L 106 point and followed after studies. The....

b) L 223 place the scales on each of the axes of Figure 3.

Author Response

Referee 1.

The authors thank the referee's positive and forward-looking comments.

We have merged the response to comments 1 and 2 by adding the following paragraphs to the manuscript, and figures have also been included as requested.

"The optimized structures of various ibuprofen-containing complexes showed that the interactions of functional groups near the chiral center, the methyl and carbonyl oxygen, and the methoxy moieties of the permethylated CD are crucial for chiral recognition. The effect of the chiral methyl moiety has a significantly smaller influence on the separation of the unsubstituted acids since the involvement of the acid proton in the formation of H-bonds between different oxygen species, including glucosidic ones, is the main separating force. According to textbooks, the distance between donor and acceptor atoms in H-bonds is usually 2.7-3.3 Ǻ. Typically, these distances were close to the lower limit for esters, as shown in Figure 4A. Since quantum-chemically optimized structures have the optimum energy at 0 K, much higher temperatures, ~400-500 K, are used in GC experiments. At experimental temperatures, the flexibility of the molecules strongly influences the conformations and interactions between molecules. Data from ongoing MD simulations will provide deeper insight into the formation of intermolecular interactions of non-relaxed molecular assemblies and their impact on the efficiency of chiral separation.

The hydrophobic interactions (van der Waals forces) prevail at slightly larger carbon-carbon distances, usually ~3.3-4 Ǻ. The interaction of chiral methyl groups with the O-methyl of permethylated CD was generally within this distance in the optimized structures. We found at least three different main methyl-methyl interactions between the molecules. The shortest distances between the methyl groups are in Figure 4A. The chiral methyl groups can interact:

(a) with O(2)CH3 and O(3)CH3 units of the same glucopyranoside unit;

(b) with O(2)CH3 and O(3)CH3 units of adjacent glucopyranoside units;

(c) with different glucopyranoside units of the macrocycle O(2)CH3 and O(2)CH3 or O(3)CH3 and O(3)CH3 separated by 2 or 3 units.

The interactions (shorter CH3…CH3 and H-bond distances) often moved in parallel, indicating that both H-bonds and hydrophobic interactions are essential for chiral separation. The weak hydrophobic interactions for both R- and S-ibuprofen also confirm that, not surprisingly, CO...HC bonds are significantly weaker than COOH...O H-bonds and the missing OH groups push hydrophobic interactions to the forefront. The values of the shortest distances between the weak H-bonds and the methyl groups are close to the limit, which may explain the weak chiral selectivity. In these cases, MD simulations will help to understand the effect of higher experimental temperatures than the optimized structures.

Analysis of the geometric center distances of permethylated CD and ibuprofen derivatives showed that longer straight-chain esters move the entire guest out of the confined state, mainly due to the interaction of the ester alkyl chain with the methoxy groups. Nevertheless, this analysis is less sensitive because the movement of the geometric centers of the guest rarely follows the same direction in different complexes. The shortest-branched alkyl, the i-propyl moiety, was closest to the host center, and the i-pentyl groups immersed nearly the same as n-butyl or n-butyl. The longest distance among the i-alkyl esters studied was in the case of i-butyl, showing interactions between the two terminal methyl moieties and the methoxy groups of the macrocycle (Figure 4B). The alkyl groups in the reversed guest orientation can more easily enter the CD cavity (not shown), albeit the host-guest interaction with the primary OCH3 groups further complicates the molecular energies. "

Reviewer 2 Report

Comments and Suggestions for Authors

Please, see the attached document.

Comments on the Quality of English Language

Minor editing of English language required.

Author Response

The authors thank the referee's positive and forward-looking comments. 

Here we would like to give our answers:

  1. We have corrected Page 4, L153.
  2. On P4 we gave a short definition of chiral selectivity, and also gave a reference.
  3. On P5 we gave a reference for the calculation of boiling points.
  4. On P5, Figure 2 we replaced the requested names.
  5. On P5 L207 we replaced the requested phrase.
  6. The Y axis is not physical values, they are relative intensity (arbitrary values), therefore, the Y axis is not written in the chromatograms. We have chosen the more decorative chromatograms and the 3a and 3b were those from the different series.